# Suppressing Overestimation in Q-Learning through Adversarial Behaviors

## Abstract

The goal of this paper is to propose a new Q-learning algorithm with a dummy adversarial player, which is called dummy adversarial Q-learning (DAQ), that can effectively regulate the overestimation bias in standard Q-learning. With the dummy player, the learning can be formulated as a two-player zero-sum game. The proposed DAQ unifies several Q-learning variations to control overestimation biases, such as maxmin Q-learning and minmax Q-learning (proposed in this paper) in a single framework. The proposed DAQ is a simple but effective way to suppress the overestimation bias thourgh dummy adversarial behaviors and can be easily applied to off-the-shelf reinforcement learning algorithms to improve the performances. A finite-time convergence of DAQ is analyzed from an integrated perspective by adapting an adversarial Q-learning. The performance of the suggested DAQ is empirically demonstrated under various benchmark environments.

## 1 Introduction

Q-learning (Watkins & Dayan, 1992) stands as one of the most foundational and widely-adopted reinforcement learning algorithms (Sutton & Barto, 2018) and has been the subject of extensive study in several works (Jaakkola et al., 1994; Tsitsiklis, 1994; Borkar & Meyn, 2000; Zou et al., 2019; Qu & Wierman, 2020; Xiong et al., 2020; Lee, 2023). Although Q-learning is known to converge to the optimal solution, it has been reported in Thrun & Schwartz (1993) that the intermediate iterates of the algorithm may suffer from maximization bias during the learning process, due to the presence of the max operator in its update formulations. Notably, the maximization bias is especially pronounced when the number of actions at a given state is substantial. To mitigate this bias, Double Q-learning was introduced in Hasselt (2010). This approach employs dual estimators and eliminates the max operator in the update, using an effective combination of the two estimators. More recently, the so-called maxmin Q-learning was proposed in Lan et al. (2020) as a means to mitigate overestimation bias. This approach introduces multiple Q-estimators and incorporates the min operator to counterbalance the bias induced by the max operator. A similar concept has also been presented in Fujimoto et al. (2018) to diminish overestimation bias in actor-critic algorithms. This is achieved by employing double Q-estimators and incorporating the min operator between the two estimators to counteract the inherent bias.

The main objective of this paper is to introduce a novel variant of Q-learning – termed Dummy Adversarial Q-learning (DAQ). This variant incorporates an additional dummy adversarial player and offers a unified framework that encompasses existing approaches. Specifically, it includes maxmin Q-learning, as proposed in Lan et al. (2020), and minmax Q-learning, introduced in this paper as a minor variant of the maxmin version, as special cases within this framework. Beyond this, the framework affords a unified perspective, aiding in the comprehension of the existing maxmin and minmax Q-learning algorithms and facilitating the analysis of their finite-time convergence in a unified fashion.

Specifically, DAQ deploys multiple Q-estimators akin to maxmin Q-learning (Lan et al., 2020), yet with additional modifications to the reward function involving the integration of constant shifts, all while preserving the original objective. The core intuition in DAQ stems from the notion that it can be interpreted as introducing a dummy player within the intrinsic Markov decision process, thus establishing a two-player zero-sum Markov game (Shapley, 1953). The dummy player strives to minimize the return without influencing the environment, which subsequently enables effective

regulation of overestimation biases in the Q update formulations. Consequently, it has been established that DAQ is equivalent to Minimax Q-learning (Littman, 1994), an algorithm formulated for zero-sum two-player Markov games (Shapley, 1953). Therefore, its convergence can be analyzed by directly applying existing analytical frameworks such as Littman & Szepesvári (1996) and Lee (2023) with minimal alterations, which are also included in this paper.

We anticipate that the principles in DAQ can be integrated into existing value-based reinforcement learning algorithms, potentially enhancing their performance. Finally, we furnish comprehensive empirical experiments to validate the effectiveness of the proposed DAQ across diverse benchmark tasks.

## 1.1 RELATED WORKS

To mitigate overestimation bias, several approaches have been introduced. Double Q-learning, as described in Hasselt (2010), utilizes dual estimators and removes the max operator in the update, employing an effective blend of the two estimators. maxmin Q-learning presented in Lan et al. (2020), introduces multiple Q-estimators and integrates the min operator to counteract the bias resulting from the max operator. A concept akin to this has been developed in Fujimoto et al. (2018), which involves the use of double Q-estimators and the incorporation of the min operator between the two estimators to neutralize the inherent bias. Bias-corrected Q-learning is proposed in Lee et al. (2013), applying intricate bias-correction terms. Weighted double Q-learning (Zhang et al., 2017) manages a weight between a single estimate and double estimates. Moreover, it was reported in Ren et al. (2021) that Double Q-learning might introduce underestimation biases, potentially leading to multiple non-optimal fixed points.

## 2 PRELIMINARIES

### 2.1 MARKOV DECISION PROCESS

For reference, we first briefly introduce the standard Markov decision problem (MDP) (Puterman, 2014), where a decision-making agent sequentially takes actions to maximize cumulative discounted rewards in environments called the Markov decision process. A Markov decision process is a mathematical model of dynamical systems with the state-space $\mathcal{S} := \{1, 2, \ldots, |\mathcal{S}|\}$ and action-space $\mathcal{A} := \{1, 2, \ldots, |\mathcal{A}|\}$. The decision maker selects an action $a$ with the current state $s$, then the state transits to a state $s'$ with probability $P(s, a, s')$, and the transition incurs a reward $r(s, a, s')$, where $P(s, a, s')$ is the state transition probability from the current state $s \in \mathcal{S}$ to the next state $s' \in \mathcal{S}$ under action $a \in \mathcal{A}$, and $r(s, a, s')$ is the reward function. At time $t \geq 0$, let $s = s_t, a = a_t$ and $s' = s_{t+1}$. For convenience, we consider a deterministic reward function and simply write $r(s_t, a_t, s_{t+1}) =: r_t, t \in \{0, 1, \ldots\}$. A (deterministic) policy $\pi : \mathcal{S} \to \mathcal{A}$ maps a state $s \in \mathcal{S}$ to an action $\pi(s) \in \mathcal{A}$. The objective of the Markov decision problem (MDP) is to find a (deterministic) optimal policy $\pi^*$ such that the cumulative discounted rewards over infinite time horizons is maximized, i.e., $\pi^* := \arg\max_{\pi \in \Theta} \mathbb{E}\left[\sum_{t=0}^{\infty} \gamma^t r_t \mid \pi\right]$, where $\gamma \in [0, 1)$ is the discount factor and $\Theta$ is the set of all admissible deterministic policies. The Q-function under policy $\pi$ is defined as $Q^\pi(s, a) = \mathbb{E}\left[\sum_{t=0}^{\infty} \gamma^t r_t \mid s_0 = s, a_0 = a, \pi\right]$, and the optimal Q-function is defined as $Q^*(s, a) = Q^{\pi^*}(s, a)$ for all $s \in \mathcal{S}, a \in \mathcal{A}$. Once $Q^*$ is known, then an optimal policy can be retrieved by the greedy policy $\pi^*(s) = \arg\max_{a \in \mathcal{A}} Q^*(s, a)$. Q-learning (Watkins & Dayan, 1992) is one of the most fundamental and popular reinforcement learning algorithms (Sutton & Barto, 2018) to solve MDP, whose Q-estimator update defined by

$$Q(s_t, a_t) \leftarrow Q(s_t, a_t) + \alpha\left[r(s_t, a_t, s_{t+1}) + \gamma\max_{a' \in \mathcal{A}} Q(s_{t+1}, a') - Q(s_t, a_t)\right] \quad (1)$$

where $\alpha > 0$ is a step-size. The Q-estimator $Q$ is known to converge to the optimal Q-function $Q^*$, by which optimal policy $\pi^*$ can be retrieved.

### 2.2 TWO-PLAYER ZERO-SUM MARKOV GAME

In a two-player zero-sum Markov game (Shapley, 1953), two agents compete with each other to maximize and minimize a return, respectively. Hereafter, these two agents will be called the user and the adversary. The user's goal is to maximize the return, while that of the adversary is to hinder

the user and minimize the return. We focus on an alternating two-player Markov game, where the user and the adversary take turns selecting their actions. The user takes an action without knowledge of the adversary. After that, the adversary observes the user's action and then takes its action.

Additional to the state space $\mathcal{S}$ and the user's action space $\mathcal{A}$, we define the adversary's action space $\mathcal{B} := \{1, 2, \ldots, |\mathcal{B}|\}$. After the user takes action $a \in \mathcal{A}$ in state $s \in \mathcal{S}$, the adversary observes $s$ and $a$, then takes action $b \in \mathcal{B}$. The state transits to $s'$ after the adversary's turn, and yields a reward based on the reward function $r(s, a, b, s')$. For the user's policy $\pi : \mathcal{S} \to \mathcal{A}$ and the adversary's policy $\mu : \mathcal{S} \times \mathcal{A} \to \mathcal{B}$, it is known that there exists an optimal policy for both user and adversary.

The seminal work by Littman (1994) introduced the following Minimax Q-learning, a Q-learning algorithm designed for zero-sum two-player Markov games:

$$Q(s_t, a_t, b_t) \leftarrow Q(s_t, a_t, b_t) + \alpha \left[ r(s_t, a_t, s_{t+1}) + \gamma \max_{a \in \mathcal{A}} \min_{b \in \mathcal{B}} Q(s_{t+1}, a, b) - Q(s_t, a_t, b_t) \right],$$

where the actions of the user and adversary $a_t, b_t$ are selected following some behavior policies. Subsequently, Littman & Szepesvári (1996) established the asymptotic convergence of Minimax Q-learning towards the optimal value derived from game theory. Recently, Lee (2023) proposed a finite-time convergence analysis of Minimax Q-learning based on the switching system model developed in Lee et al. (2023). Later in this paper, we provide a new perspective that the proposed Q-learning algorithms can be analyzed based on Minimax Q-learning for solving two-player zero-sum Markov games.

## 3 Controlling Overestimation Biases

In this section, we briefly overview some existing algorithms that are designed to neutralize the overestimation biases inherent in standard Q-learning.

### 3.1 Maxmin Q-Learning

To control overestimation biases in the naïve Q-learning, maxmin Q-learning was suggested in Lan et al. (2020), which maintains $N$ estimates of the Q-functions, $Q_i$, and using the minimum of these estimates for the Q-learning target. The corresponding update is given by

$$Q_i(s_t, a_t) \leftarrow Q_i(s_t, a_t) + \alpha \left[ r(s_t, a_t, s_{t+1}) + \gamma \max_{a' \in \mathcal{A}} \min_{j \in \{1, \ldots, N\}} Q_j(s_{t+1}, a') - Q_i(s_t, a_t) \right]$$

for some randomly selected $i$. When $N = 1$, the naïve Q-learning is recovered. As $N$ increases, the updates tend to suppress the overestimation biases.

### 3.2 Double Q-Learning

A double estimator was first introduced in Double Q-learning (Hasselt, 2010) to reduce the overestimation of Q-learning induced by the max operator. The update is given as follows:

$$Q_i(s_t, a_t) \leftarrow Q_i(s_t, a_t) + \alpha \left[ r(s_t, a_t, s_{t+1}) + \gamma Q_j(s_{t+1}, \arg\max_{a' \in \mathcal{A}} Q_i(s_{t+1}, a')) - Q_i(s_t, a_t) \right]$$

where $i \in \{1, 2\}$ is randomly selected, and $j = 1$ if $i = 2$ and $j = 2$ when $i = 1$.

### 3.3 Twin-Delayed Deep Deterministic Policy Gradient (TD3)

Twin-delayed deep deterministic policy gradient (TD3) (Fujimoto et al., 2018) is an improved actor-critic algorithm, which considered a target that takes the minimum between the two Q-estimates: $Y^{target} = r(s, a, s') + \gamma \min_{j \in \{1,2\}} Q_{\theta'_j}(s', \pi_{\phi_i}(s'))$ for a pair of actors $\pi_{\phi_i}$ and critics $Q_{\theta_i}$, and targets $\pi_{\phi'}$ and $Q_{\theta'}$. This modification leads to reduced overestimation biases due to the newly introduced min operator.

# 4 PROPOSED ALGORITHMS

## 4.1 MINMAX Q-LEARNING

First, we introduce a minor modification to maxmin Q-learning, accomplished by simply switching the order of the min and max operators. In particular, consider the following update:

$$Q_i(s_t, a_t) \leftarrow Q_i(s_t, a_t) + \alpha \left[ r(s_t, a_t, s_{t+1}) + \gamma \min_{j \in \{1,\dots,N\}} \max_{a' \in \mathcal{A}} Q_j(s_{t+1}, a') - Q_i(s_t, a_t) \right]$$

for all $i \in \{1, \dots, N\}$ (synchronous case) or randomly selected $i$ (asynchronous case), where $N$ represents the number of Q-estimators, the selection of which can be flexibly adapted depending on the problem. We call this algorithm minmax Q-learning[1]. Although it represents a subtle modification of maxmin Q-learning, its characteristics have remained unexplored in existing literature until now. In this paper, we provide experimental results demonstrating that the proposed minmax Q-learning can also effectively regulate overestimation biases. Moreover, we will soon demonstrate that minmax Q-learning can be integrated into a broader class of algorithms, named the Dummy Adversarial Q-learning (DAQ), which will be introduced in the following subsection. This integration enables a unified analysis and interpretation framework alongside maxmin Q-learning.

## 4.2 DUMMY ADVERSARIAL Q-LEARNING (DAQ)

In this paper, we introduce a novel Q-learning algorithm, termed Dummy Adversarial Q-learning (DAQ), designed to exert more effective control over the overestimation biases induced by the max operator. The algorithms are presented in two versions: the maxmin and the minmax versions, and the corresponding Q-updates are summarized as follows.

**Dummy Adversarial Q-learning (maxmin version)**

$$Q_i(s_t, a_t) \leftarrow Q_i(s_t, a_t) + \alpha \left[ r(s_t, a_t, s_{t+1}) + b_i + \gamma \max_{a' \in \mathcal{A}} \min_{j \in \{1,\dots,N\}} Q_j(s_{t+1}, a') - Q_i(s_t, a_t) \right]$$
(2)

for all $i \in \{1, \dots, N\}$ (synchronous case) or randomly selected $i$ (asynchronous case).

**Dummy Adversarial Q-learning (minmax version)**

$$Q_i(s_t, a_t) \leftarrow Q_i(s_t, a_t) + \alpha \left[ r(s_t, a_t, s_{t+1}) + b_i + \gamma \min_{j \in \{1,\dots,N\}} \max_{a' \in \mathcal{A}} Q_j(s_{t+1}, a') - Q_i(s_t, a_t) \right]$$
(3)

for all $i \in \{1, \dots, N\}$ (synchronous case) or randomly selected $i$ (asynchronous case).

The main difference from the maxmin and minmax Q-learning algorithms lies in the modified reward, $r(s_t, a_t, s_{t+1}) + b_i$, where the constant $b_i$ depending on the Q-estimator index $i$ is added to the original reward when updating the target[2]. The constant shift terms $b_i$ can be either positive or negative, and either random or deterministic, contingent upon the environment, functioning as design parameters. In subsequent sections, experimental results are presented to illustrate the impacts of different choices of $b_i$. Due to the constants, each estimator $Q_i$ learns a modified optimal Q-function

$$Q_i^*(s, a) := \mathbb{E} \left[ \sum_{t=0}^{\infty} \gamma^t (r(s_t, a_t, s_{t+1}) + b_i) \,\middle|\, s_0 = s, a_0 = s, \pi^* \right]$$

which differs from the original optimal Q-function

$$Q^*(s, a) := \mathbb{E} \left[ \sum_{t=0}^{\infty} \gamma^t r(s_t, a_t, s_{t+1}) \,\middle|\, s_0 = s, a_0 = s, \pi^* \right]$$

---

[1]Note that our minmax Q-learning is different from Minimax Q-learning (Littman, 1994) which was designed for a two-player Markov game.

[2]Unlike conventional reward shaping methods using shaping function $F(s, a, s')$ which depends on state and action (Ng et al., 1999; Wiewiora et al., 2003), here we use a fixed constant.

with the constant bias $\sum_{t=0}^{\infty} \gamma^t \mathbb{E}[b_i] = \frac{\mathbb{E}[b_i]}{1-\gamma}$, i.e., $Q_i^*(s,a) - \frac{\mathbb{E}[b_i]}{1-\gamma} = Q^*(s,a)$. Recalling that our goal is to find an optimal policy and finding the optimal Q-function aids this, adding a constant to the reward function does not affect the optimal policy which comes from a relative comparison among Q-function values of the given state, i.e., $\pi^*(s) = \arg\max_{a \in \mathcal{A}} Q^*(s,a) = \arg\max_{a \in \mathcal{A}} Q_i^*(s,a), i \in \{1,2,\ldots,N\}$.

A larger $N$ might lead to slower convergence but has the potential to significantly mitigate overestimation biases. The reduction in convergence speed can be counterbalanced by a synchronized update of the Q-iterate, a topic that will be elaborated upon in the ensuing subsection.

In summary, DAQ is capable of learning an optimal policy through the utilization of multiple Q-estimators without modifying the original objective. However, the adjustments made to the rewards and updates have the potential to regulate the overestimation biases arising from the max operator.

The fundamental intuition behind DAQ hinges on the viewpoint that it can be regarded as introducing a dummy player to the inherent Markov decision process, thereby establishing a two-player zero-sum Markov game (Shapley, 1953). Consequently, DAQ can be interpreted as Minimax Q-learning (Littman, 1994) for the Markov game, featuring a dummy player who, while not influencing the environment, can regulate the overestimation biases. These insights will be further detailed in the subsequent sections. Lastly, we foresee the extension of this approach to deep Q-learning and actor-critic algorithms as a viable means to augment their performances.

### 4.3 Discussion on Asynchronous versus Synchronous Updates

There exist two potential implementations of DAQ: synchronous updates and asynchronous updates. In asynchronous updates, each estimator $i$ is randomly selected from some distribution $\mu$. Conversely, in synchronous updates, all estimators $i \in \{1,2,\ldots,N\}$ are updated at every iteration. A trade-off emerges between convergence speed and computational cost per iteration step in these versions. The asynchronous version converges more slowly than the synchronous version, as it updates only one estimator at a time, while the latter updates all estimators. However, the computational cost per step is higher in the synchronous version. Note that in the synchronous version, with $b_i = 0$ (in the case of minmax or maxmin Q-learning algorithms), all the Q-estimators will have identical values at every step, provided the initial Q-estimators are not set randomly. Therefore, to implement the synchronous version, one should either initialize the multiple Q-estimators randomly or apply nonzero shifts $b_i$, ideally choosing these shifts at random.

### 4.4 Interpretation from the Two-Player Zero-Sum Game

DAQ can be interpreted as the Minimax Q-learning algorithm for a two-player zero-sum game (Lee, 2023; Shapley, 1953; Littman, 1994). The index $i$ of equation 2 and equation 3 can be interpreted as the action selected by the adversary, $r(s_t, a_t, s_{t+1}) + b_i$ is the reward affected by both the user and adversary, and $\mathcal{B} = \{1,2,\ldots,N\}$ is the action set of the adversary.

More specifically, consider the addition of a dummy player who, at each stage, executes an action $i \in \mathcal{B}$, incorporating the additional value $b_i$ into the reward $r(s_t, a_t, s_{t+1})$; that is, $r(s_t, a_t, s_{t+1})$ becomes $r(s_t, a_t, s_{t+1}) + b_i$. This player endeavors to minimize the return, the discounted summation of rewards, signifying its adversarial role, and thus reformulating the MDP into a two-player zero-sum game. The dummy player's action does not alter the inherent transition probabilities of the MDP, preserving the user's ability to attain its goal unimpeded. However, it can modify the overestimation biases during the learning updates, serving as the core concept of DAQ. The Minimax Q-learning algorithm, which is suitable for a two-player zero-sum game, can be adapted to this redefined game framework leading to

$$Q(s_t, a_t, i) \leftarrow Q(s_t, a_t, i) + \alpha \left[ \underbrace{r(s_t, a_t, i)}_{=r_t + b_i} + \gamma \max_{a \in \mathcal{A}} \min_{j \in \{1,\ldots,N\}} Q(s_{t+1}, a, j) - Q(s_t, a_t, i) \right]$$

and

$$Q(s_t, a_t, i) \leftarrow Q(s_t, a_t, i) + \alpha \left[ \underbrace{r(s_t, a_t, i)}_{=r_t + b_i} + \gamma \min_{j \in \{1,\ldots,N\}} \max_{a \in \mathcal{A}} Q(s_{t+1}, a, j) - Q(s_t, a_t, i) \right]$$

These algorithms are exactly identical to DAQ (minmax and maxmin versions) in this paper. Since the adversary tries to minimize the reward, she or he may cause underestimation biases, regulating the overestimation biases in the naïve Q-learning. We observe that the adversarial actions have no impact on the user's goal as these actions are artificially incorporated into both the reward and the Q-updates, rendering them redundant. However, the incorporation of such dummy adversarial behavior can alter the overestimation biases.

Moreover, the sole distinction between the minmax and maxmin versions in DAQ lies in the reversed order of alternating turns. Specifically, within the maxmin version, the adversary holds more advantages over the user as it can observe the actions undertaken by the user and respond accordingly. This suggests that the algorithm tends to have underestimation biases. In contrast, within the minmax version, the user maintains more advantages over the adversary, afforded by the ability to observe the adversary's actions. This means that the algorithm is inclined to have overestimation biases.

## 4.5 FINITE-TIME CONVERGENCE ANALYSIS

Since DAQ can be interpreted as Minimax Q-learning for the two-player Markov game, existing analyses such as Lee (2023) and Littman & Szepesvári (1996) can be directly applied with minimal modifications. In this paper, we refer to the simple finite-time analysis presented in Lee (2023).

For simplicity of analysis, let us suppose that the step-size is a constant $\alpha \in (0, 1)$, the initial iterate $Q$ satisfies $\|Q\|_\infty \leq 1$, $|r(s, a, s')| \leq 1$ for all $(s, a, s') \in \mathcal{S} \times \mathcal{A} \times \mathcal{S}$, and $\{(s_k, a_k, s'_k)\}_{k=0}^\infty$ are i.i.d. samples under the behavior policy $\beta$ for action $a \in \mathcal{A}$, where the behavior policy denotes the strategy by which the agent operates to collect experiences. Moreover, we assume that the state at each time is sampled from the stationary state distribution $p$, and in this case, the state-action distribution at each time is identically given by

$$d(s, a) = p(s)\beta(a|s) \quad \forall (s, a, b) \in \mathcal{S}.$$

Moreover, let us suppose that $d(s, a) > 0$ holds for all $s \in \mathcal{S}, a \in \mathcal{A}$. In the asynchronous version, the distribution $\mu$ of $i$ can be seen as the adversary's behavior policy. We assume also that $\mu(i) > 0, \forall i \in \{1, 2, \ldots, N\}$. Moreover, let us define $d_{\max} := \max_{(s,a,i) \in \mathcal{S} \times \mathcal{A} \times \{1,2,\ldots,N\}} d(s, a)\mu(i) \in (0, 1)$, $d_{\min} := \min_{(s,a,i) \in \mathcal{S} \times \mathcal{A} \times \{1,2,\ldots,N\}} d(s, a)\mu(i) \in (0, 1)$, and $\rho := 1 - \alpha d_{\min}(1 - \gamma)$. Applying Theorem 4 from Lee (2023) directly yields the following theorem.

**Theorem 1** *Let us consider the asynchronous version of DAQ. For any $t \geq 0$, we have*

$$\mathbb{E}[\|Q_i - Q_i^*\|_\infty] \leq \frac{27d_{\max}|\mathcal{S} \times \mathcal{A}|N\alpha^{1/2}}{d_{\min^{3/2}}(1 - \gamma)^{5/2}} + \frac{6|\mathcal{S} \times \mathcal{A}|^{3/2}N^{3/2}}{1 - \gamma}\rho^t$$

$$+ \frac{24\gamma d_{\max}|\mathcal{S} \times \mathcal{A}|^{2/3}N^{2/3}}{1 - \gamma}\frac{3}{d_{\min}(1 - \gamma)}\rho^{t/2-1},$$

*where $Q_i$ is the $i$-th estimate at iteration step $t$, and $Q_i^*$ is the optimal Q-function corresponding to the $i$-th estimate.*

Note that in the bound detailed in Theorem 1, both the second and third terms diminish exponentially as the iteration step $t$ approaches infinity. The first term stands as a constant bias term, originating from the use of the constant step-size $\alpha$, but this term disappears as $\alpha \to 0$. This suggests that by employing a sufficiently small step-size, the constant bias can be effectively controlled. Along similar lines, the finite-time error bound for the synchronous version can be derived as in Theorem 1 with $d_{\max}$ and $d_{\min}$ replaced with $d_{\max} := \max_{(s,a,i) \in \mathcal{S} \times \mathcal{A}} d(s, a) \in (0, 1)$ and $d_{\min} := \min_{(s,a,i) \in \mathcal{S} \times \mathcal{A}} d(s, a) \in (0, 1)$, respectively. Note also that in this case, the error bound in Theorem 1 becomes smaller because $d_{\min}$ tends to increase and $d_{\max}$ tends to decrease in the synchronous case.

## 5 EXPERIMENTS

In this section, the efficacy of DAQ is demonstrated across several benchmark environments, and a comparative analysis is provided between DAQ and existing algorithms, including standard Q-learning, Double Q-learning, maxmin Q-learning, and minmax Q-learning. An $\epsilon$-greedy behavior

based on $\sum Q_i$ and tabular action-values initialized with 0 are used. Any ties in $\epsilon$-greedy action selection are broken randomly.

## 5.1 MDP ENVIRONMENTS

**Grid World** in Hasselt (2010) is considered, which is illustrated in Figure 1a. In this environment, the agent can take one of four actions at each state: move up, down, left, or right. The episode starts at S, and the goal state is G. For each transition to non-terminal states, the agent receives a random reward $r = -12$ or $r = +10$ with equal probabilities. When transiting to the goal state, every action incurs $r = +5$, and the episode is terminated. If the agent follows an optimal policy, the episode is terminated after five successive actions, and hence, the expected reward per step is $+0.2$. We will also consider another reward function adapted from Wang et al. (2020). For each transition, the agent receives a fixed reward $r = -1$. When transiting to the goal state, every action yields $r = -35$ or $r = +45$ with equal probabilities, and the episode ends. Under an optimal policy, the expected average reward per step is also $+0.2$.

**Sutton's Example** shown Figure 6.5 of Sutton & Barto (2018) is illustrated in Figure 1b. There exist two non-terminal states A and B at which two actions, left and right, are admitted. There are two terminal states denoted with grey boxes and A is the initial state. Executing the action right leads to the termination of an episode with no rewards, while the action left leads to state B with no reward. There exist many actions at state B, which will all immediately terminate the episode with a reward drawn from a normal distribution $\mathcal{N}(-0.1, 1)$. In our experiments, we will consider the two cases $r \sim \mathcal{N}(\mu, 1)$ with $\mu = -0.1$ and $\mu = +0.1$.

**Weng's Example** from Weng et al. (2020) is adapted, and it is illustrated in Figure 1c. There exist $M + 1$ states labeled as $\{0, \ldots, M\}$ with two actions, left and right, at each state, where 0 is the initial state. Choosing action right at the initial state leads to the terminal state, while choosing left leads to random transitions to one of the other $M$ states. At the initial state, both actions result in the reward $r = 0$. At any $s \in \{1, 2, \ldots, M\}$, it transits to 0 with the action right. On the other hand, the episode is terminated with the action left. Both actions incur a reward drawn from a normal distribution $\mathcal{N}(-0.1, 1)$.

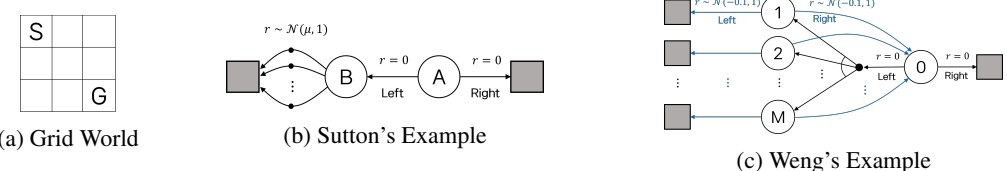

(a) Grid World    (b) Sutton's Example    (c) Weng's Example

Figure 1: MDP Environments

## 5.2 EMPIRICAL RESULTS

**Grid World** For exploration, we employ the $\epsilon$-greedy policy with a count-based $\epsilon(s) = 1/\sqrt{n(s)}$, where $n(s)$ represents the number of visits to state $s$. The step-size is designated as $\alpha(s, a) = 1/n(s, a)^{0.8}$, where $n(s, a)$ denotes the number of visits[3] to the state-action pair $(s, a)$.

The left subfigure in Figure 2 shows a comparative analysis of several algorithms. Using Hasselt's reward function from Hasselt (2010), an algorithm is supposed to underestimate the value of the non-terminal transitions to reach the goal. Q-learning tends to prioritize exploration over reaching the goal due to its overestimation, which results in the average reward per step remaining around $-1$. Minmax Q-learning demonstrates superior performance compared to Q-learning, as its average reward per step tends to converge to $0.2$. With appropriate reward shifts, the minmax version of DAQ markedly enhances the performance of minmax Q-learning, as illustrated by the progression from the blue curve (minmax Q-learning) to the purple (DAQ minmax version). Here, we set $b_1 = -5$ and $b_2 = -10$ for both DAQs, which were determined based on some experiments in Figure 6a in the supplemental document.

---

[3] If double estimators $Q_1$ and $Q_2$ are employed with asynchronous updates, then each estimator $Q_i$ must be equipped with its own counter $n_i(s, a)$ that records the number of visits to the pair $(s, a)$ when $Q_i$ is updated.

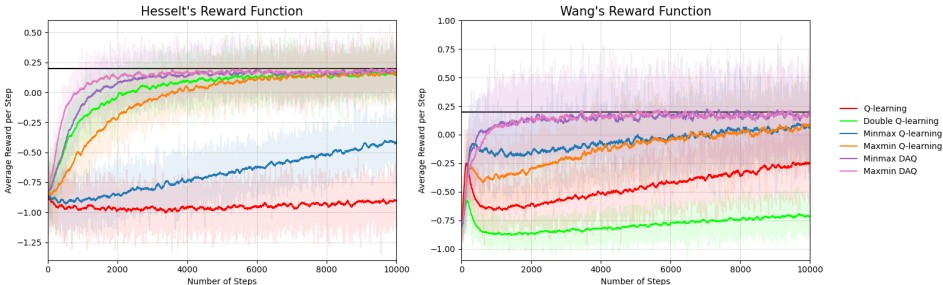

Figure 2: Experiments with grid world environment, where $\gamma = 0.95$ were employed. Learning curves are averaged over 10,000 experiments, and the moving averages with a window size of 100 are shown in the vivid lines. In Figure 5 in the supplement, experiments were conducted with different parameter values, $\alpha = 0.1$ or $\epsilon = 0.1$, where the trends were similar.

The right subfigure in Figure 2 depicts results corresponding to Wang's reward function in Wang et al. (2020), where overestimation is more beneficial for reaching the goal state. Due to its inherent bias towards underestimation corresponding to the goal state, Double Q-learning persistently explores, showcasing the slowest convergence to the optimal average reward per step. Within the non-reward-shifting models, minmax Q-learning provides the best performance. This suggests that in certain environment setups, the minmax Q-learning holds inherent advantages over the maxmin Q-learning. The forementioned non-DAQ models all converge to suboptimal values for the averaged reward per step in their early stages of evolution. This implies that, in the scenarios with a constant step-size, they possess larger biases in their error bounds. Conversely, in the decreasing step-size scenario, this means a slower rate of convergence. However, by implementing the negative reward shifts $b_1 = -5$ and $b_2 = -10$, both minmax and maxmin versions of DAQ exhibit accelerated convergence to the optimal average reward per step.

**Sutton's Example[4] with $\mu = -0.1$** Since $\mu$ is less than $0$, the optimal action at state A is right. Consequently, 5% is the optimal ratio, illustrated as the black horizontal lines. Figure 3 depicts the learning curves of several algorithms. The first, second, and last subfigures display results associated with 8, 20, and 100 actions at state B, respectively. On the first subfigure with 8 actions, with a constant step-size $\alpha = 0.1$, all the discussed algorithms converge to the optimal ratio. However, DAQs attain precisely a 5% ratio relatively early on, around the 180-th episode, outperforming Double Q-learning which continues to exhibit biases at this stage.

The second subfigure illustrates learning curves corresponding to 20 actions, that are similar to the results in the first subfigure. The DAQ versions exhibit a delay in learning the optimal action, maintaining a 50% ratio for a duration, but eventually, they do converge to the optimal 5% ratio at a faster rate than Double Q-learning. Here, minmax Q-learning exposes a drawback: it exhibits a markedly slow learning pace, especially as the action space expands.

The final subfigure depicts the learning curves associated with 100 actions. As expected, minmax Q-learning learns at a slower pace; therefore, we have omitted its graph for clarity. When negative shifts are implemented in the reward function, the related DAQ versions converge to the optimal 5% ratio more swiftly compared to both Q-learning and maxmin Q-learning but at a more gradual pace than the previous cases. To expedite convergence, synchronous DAQ was also tested. A notable enhancement in the learning curve is evident transitioning from the purple curve to the olive one. In the synchronized DAQ, it learns to take right action quickly until around 250-th episode. We additionally conducted tests with decreasing $\alpha_n = \frac{10}{n+100}$, and the learning curves closely resemble those in the constant step-size case, detailed in Figure 9 in the supplement.

**Weng's Example** In this experiment, we utilize $M \in \{8, 20, 100\}$, with results displayed in Figure 4. Note that in Sutton's Example, the action space enlarges, whereas in Weng's Example, it's the state space that expands as $M$ increases. Since every action from state 1 to M results in a negative expected reward, the optimal action at state 0 is the right action. Given $\epsilon$-greedy exploration, a 5%

---

[4]The experimental results with $\mu = +0.1$ is shown in Appendix A.2 in the supplement, and the result is not much different.

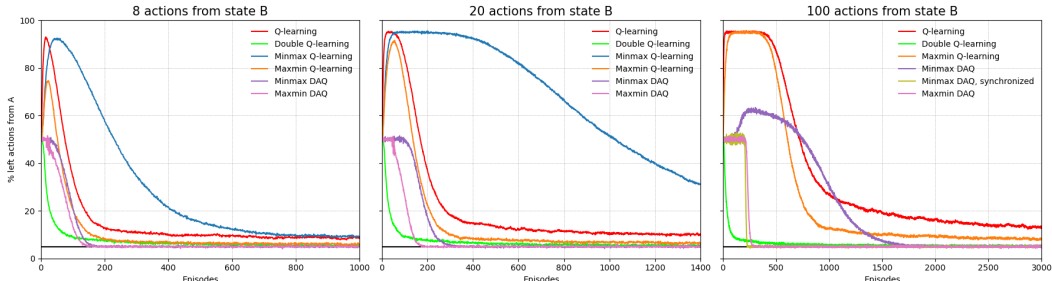

Figure 3: Experiments with Sutton's example with $\mu = -0.1$, experimented with $\epsilon = 0.1$, $\alpha = 0.1$ and $\gamma = 1$. The three subfigures show the learning curves for different numbers of actions at state B, where the learning curves were averaged over 10,000 experiments. For DAQs, $b_1 = -1$ and $b_2 = -2$ were used. In each subfigures, the number of episodes are different in order to show the convergence of the algorithms.

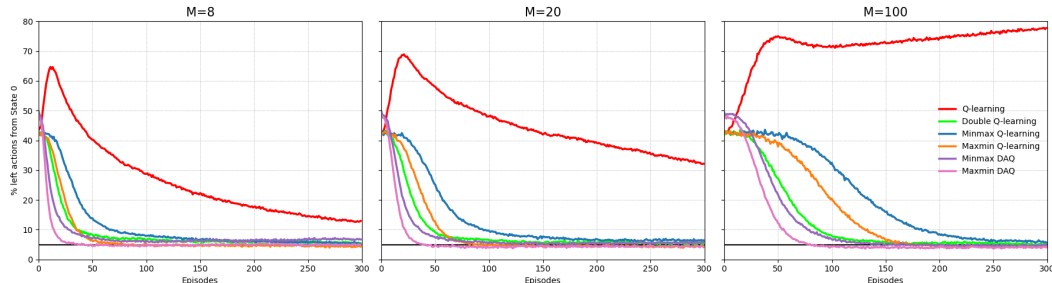

Figure 4: Experiments with Weng's example, where $\alpha_n = \frac{10}{n+100}$, $\epsilon = 0.1$ and $\gamma = 1$ are used. The three subfigures correspond to the results with different numbers of states $M$. The learning curves were averaged over 10,000 experiments. Similar results can be obtained with the constant step-size $\alpha = 0.1$ as shown in Figure 10b in the supplement.

ratio of taking the left action is optimal, represented by 5% horizontal lines. The learning curves of DAQs are highlighted with pink and purple colors, revealing that DAQ versions attain the optimal action faster compared to the other algorithms. In DAQ, $b_1 = +0.0001$ and $b_2 = +0.0002$ are incorporated into the reward function to attain better performance; these are minor compared to the reward $\mathcal{N}(-0.1, 1)$ or $0$. When the shifts get closer to zero, the DAQs converge faster as shown in Figure 10a in the supplemental document. Note that the graphs for DAQ versions begin at nearly 50%, in contrast to others starting at around 40%. With the incorporation of shifted rewards, DAQ versions achieve more uniform exploration, preventing biased exploration.

## 6 CONCLUSION

In this paper, we have introduced DAQ to mitigate overestimation biases in Q-learning. The DAQ proposed is capable of learning an optimal policy by leveraging multiple Q-estimators without altering the original objective. The fundamental intuition underpinning DAQ is based on the perspective that it can be conceptualized as introducing a dummy player to the inherent MDP, thereby formulating a two-player zero-sum game. This dummy player endeavors to minimize the return without influencing the environment, and as a result, can effectively regulate overestimation biases. DAQ successfully unifies several algorithms, including maxmin Q-learning and minmax Q-learning – the latter being a slightly modified version of the former – under a single framework. Moreover, the principles of DAQ can be integrated into existing value-based reinforcement learning algorithms. To illustrate the efficacy of DAQ algorithms, we have executed experiments across various environments. We envision the extension of this approach to deep Q-learning and actor-critic algorithms as a promising avenue to enhance their performances; this aspect remains open for further research.

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

# A    ADDITIONAL EMPIRICAL RESULTS

Here, we give more experimental results which are omitted from Section 5.2.

## A.1    GRID WORLD

The experimental results using the Grid World environment is depicted in this subsection.

In Figure 5, parameter values other than mainly suggested in Section 5.2 are tested. The first column is the same as Figure 2 for comparison, and the second and the last column use parameter values of $\alpha = 0.1$ or $\epsilon = 0.1$. Figure 5a shows the result using Hasselt's reward function. The second subfigure presents a comparative analysis featuring a decreasing step-size and $\epsilon$-greedy exploration with a constant $\epsilon = 0.1$, and the result is not much different from the first subfigure. The last subfigure provides a comparative analysis employing a constant step-size $\alpha = 0.1$ and a constant $\epsilon = 0.1$, where the DAQs continue to display superior results. Figure 5b shows the result using Wang's reward function. Both minmax Q-learning and maxmin Q-learning show significantly better results than Q-learning and Double Q-learning. In the second subfigure, the distinctions between DAQs and other considered algorithms become clearer when a constant step-size $\alpha = 0.1$ is used. Employing a constant $\epsilon = 0.1$ does not significantly alter the comparative results, as is shown in the last subfigure.

In Figure 6, shift values other than mainly suggested in Section 5.2 are tested. Figure 6a shows the result using Hasselt's rewards. For both DAQs, we empirically checked that the negative value of $b_i$ facilitates converging to optimal while the positive values hinder convergence. Performance is better with $(b_1, b_2) = (-5, -10)$ than $(b_1, b_2) = (-1, -2)$, and even slightly better with $(b_1, b_2) = (-20, -30)$. However, we decided to use $-5$ and $-10$ in Figure 2 since the difference is not much distinguishable. Figure 6b shows the results using Wang's rewards, especially when $(b_1, b_2) = (-1, -2)$ is used for DAQs at Grid World Experiment using Wang's reward function. The result with decreasing step-size is not much different from that of Figure 2, but the convergence at constant step-size was better at Figure 2 which used $(b_1, b_2) = (-5, -10)$.

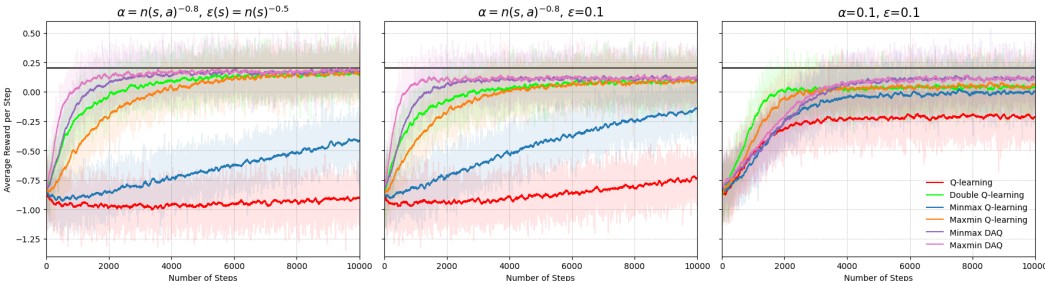

(a) Result using Hasselt's reward function. Decreasing step-size is used on the first and the second subfigures, and count-based $\epsilon(s)$ is also used on the first subfigure.

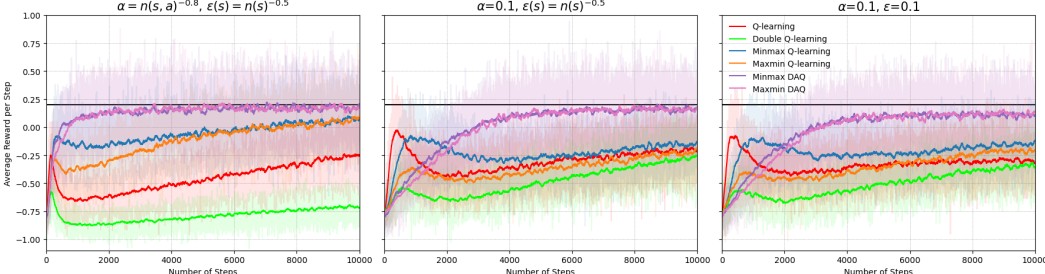

(b) Result using Wang's reward function. Decreasing step-size is used on the first subfigure, and count-based $\epsilon(s)$ is also used on the first and the second subfigures.

Figure 5: Result of Grid World Experiment. $\gamma = 0.95$ is used. Data is averaged over 10,000 experiments, and the moving average with a window size of 100 is shown in vivid lines. The horizontal lines at 0.2 represent the average reward per step for optimal policy.

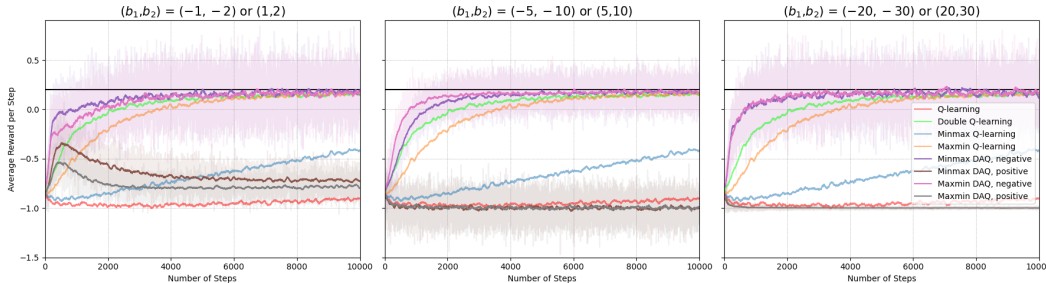

(a) Result using Hasselt's reward function with various shift values, which are presented in titles of each subfigures. $\alpha = 1/n(s,a)^{0.8}$ and count-based $\epsilon(s)$ are used. The graphs of Q-learning, Double Q-learning, minmax Q-learning and maxmin Q-learning are the same for all three subfigures for better comparison, and only their moving averages are drawn here.

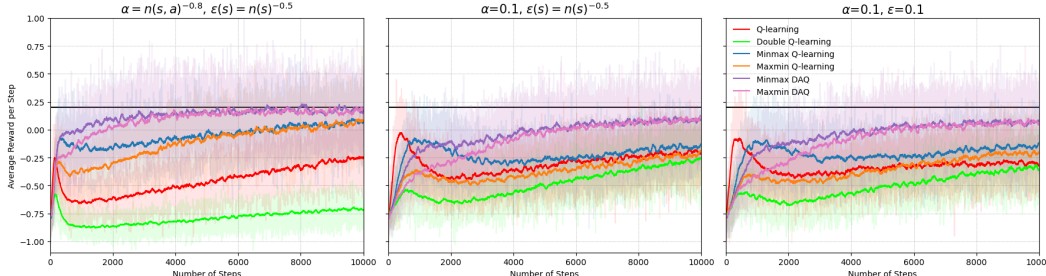

(b) Result with Wang's reward function. $\alpha = 1/n(s,a)^{0.8}$ is used on the first subfigure, and count-based $\epsilon(s)$ is also used on the first and the second.

Figure 6: Result of Grid World Experiments. $\gamma = 0.95$ is used. Data is averaged over 10,000 experiments, and the moving average with a window size of 100 is shown in vivid lines. The horizontal lines at $0.2$ represent the average reward per step for optimal policy.

## A.2 SUTTON'S EXAMPLE

**Sutton's Example with $\mu = +0.1$** In this scenario, actions at state B are anticipated to yield positive rewards, thus making left the optimal action at state A. To explore this environment, the $\epsilon$-greedy exploration with $\epsilon = 0.1$ is employed. Consequently, after the convergence of the Q iterate to $Q^*$, it is optimal to take the action left at state A 95% of the time, aligning with the logic of $\epsilon$-greedy exploration. The action right is chosen for the remaining 5%. These two ratios are illustrated by the black horizontal lines in Figure 7. Each curve other than the lines represents the evolution of the ratio of selecting left actions. In this setting, the designer has the autonomy to determine the number of available actions at state B, which, in this instance, has been set to eight.

Figure 7a illustrates the learning curves associated with the decreasing step-size, denoted as $\alpha_n = \frac{10}{n+100}$ for episode $n$. Minmax Q-learning reaches the optimal ratio considerably quicker than the other algorithms, and both Q-learning and maxmin Q-learning approach the optimum but exhibit a preference for the right action during the exploration phase. The grey curve in the graph represents the maxmin version of DAQ with $b_1 = +0.01$ and $b_2 = +0.02$, which were determined from experiments in Figure 8. A significant alteration in the learning curve can be observed when comparing the grey and the orange curves. Even though $0.01$ or $0.02$ are relatively small quantities when compared to the rewards, these reward shifts significantly aid the algorithm in learning an optimal policy effectively and efficiently.

Figure 7b displays the learning curves associated with a constant step-size, $\alpha = 0.1$. One can observe that only the DAQ versions converge to the optimal ratio, while the remaining algorithms settle at values with larger biases. DAQs with positive $b_i$ maintain exploratory actions up to approximately the 300-th episode, after which they acquire the proficiency to choose the optimal left action.

Figure 8 is another experiment with various shift values tested to Sutton's Example with $\mu = +0.1$, and the grounds for the chosen shift values at Figure 7a. Starting from positive $(b_1, b_2) = (+1, +2)$

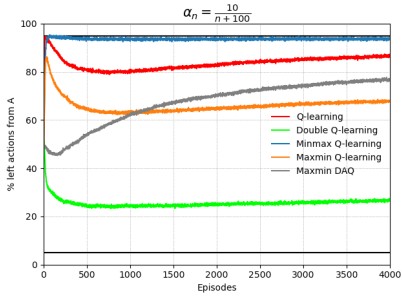 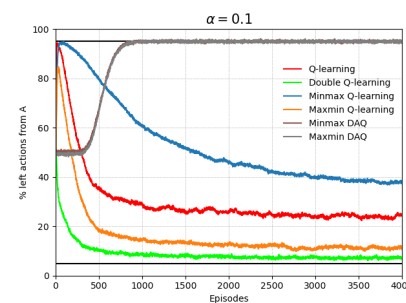

(a) Experiments using decreasing step-sizes, where minmax Q-learning exhibits the best performance. Minmax Q-learning can be considered as minmax DAQ with $b_i = 0$.

(b) Experiments with constant step-sizes, where $b_1 = +1$ and $b_2 = +2$ are used to both DAQs. Note that the curves of the two DAQs are almost overlapped.

Figure 7: Experiments with Sutton's example with $\mu = +0.1$, where 95% is optimal, $\epsilon = 0.1$ and $\gamma = 1$ were used. The two subfigures show learning curves with different step-size rules. The learning curves were averaged over 5,000 experiments.

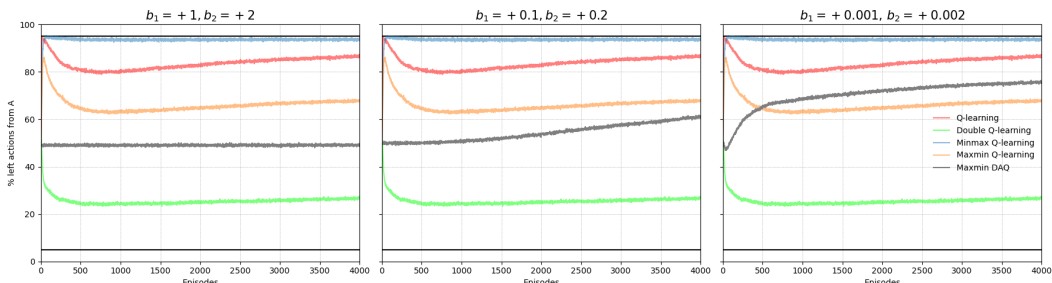

Figure 8: Result of various shift values, which are presented in titles of each subfigures, using Sutton's Example with $\mu = +0.1$. Here, 95% line is optimal and 5% is sub-optimal. $\alpha_n = \frac{10}{n+100}$, $\epsilon = 0.1$ and $\gamma = 1$ are used. Data is averaged over 5,000 experiments. The graphs of Q-learning, Double Q-learning, minmax Q-learning and maxmin Q-learning are the same for all three subfigures for better comparison.

and negative $(b_1, b_2) = (-1, -2)$, we adjusted the amount by 10 times, from $10^{-3}$ to $10^1$ scale. All positive shifts disturb the algorithm to converge: the larger the value, the slower the convergence. On the other hand, among the negative values, $10^{-2}$ is the most effective while $10^{-3}$ converges faster to the optimal at first, but is soon saturated and shows the same slope with Q-learning or maxmin Q-learning. Especially, positive shifts are shown in Figure 8. $(b_1, b_2) = (+1, +2)$ or $(+10, +20)$ cannot learn any specific policy, and the algorithm starts to converge as the amount of shift decreases.

Figure 9 shows the results using Sutton's Example with $\mu = -0.1$. The graphs are similar to Figure 3, except for the complete convergence of Q-learning and maxmin Q-learning here. Decreasing step-size makes the algorithms converge to exact optimal compared to constant step-size. We additionally suggested the graphs of synchronized DAQ for all three subfigures. When the size of the action space is small, the synchronized DAQ fluctuates a lot, though steadily gets closer to the optimal and achieves the optimal faster than any other algorithms.

## A.3 WENG'S EXAMPLE

Figure 10 shows the experimental results using Weng's Example. In Figure 10a, we give results with varying shifts. We tested with a scale from $10^1$ to $10^{-4}$, both for positive and negative values of shifts. As the value of shifts gets closer to zero, DAQs converge faster. The result with constant step-size is shown in Figure 10b. The graphs are very similar to Figure 4 except for the behavior of Q-learning.

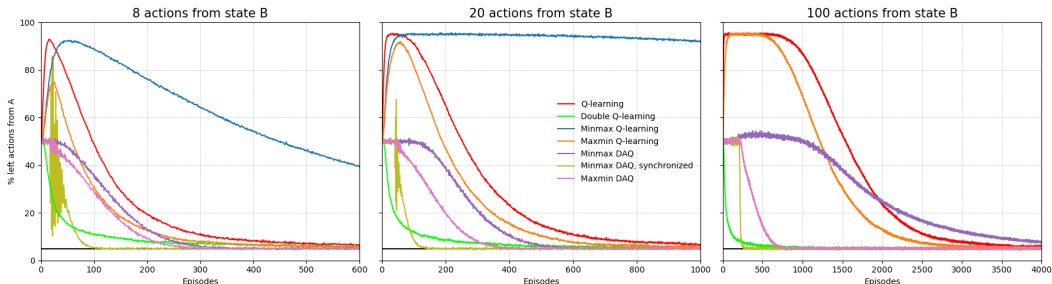

Figure 9: Result using Sutton's Example with $\mu = -0.1$, where a 5% line is optimal. From the first to the last subfigure: 8 actions, 20 actions, and 100 actions from state B, respectively. $\epsilon = 0.1$, $\alpha_n = \frac{10}{n+100}$ and $\gamma = 1$ are used. Data is averaged over 10,000 experiments. $b_1 = -1$ and $b_2 = -2$ are used for shifting. We increased the number of episodes from 600 to 1,000 and 4,000 to check the convergence of the algorithms.

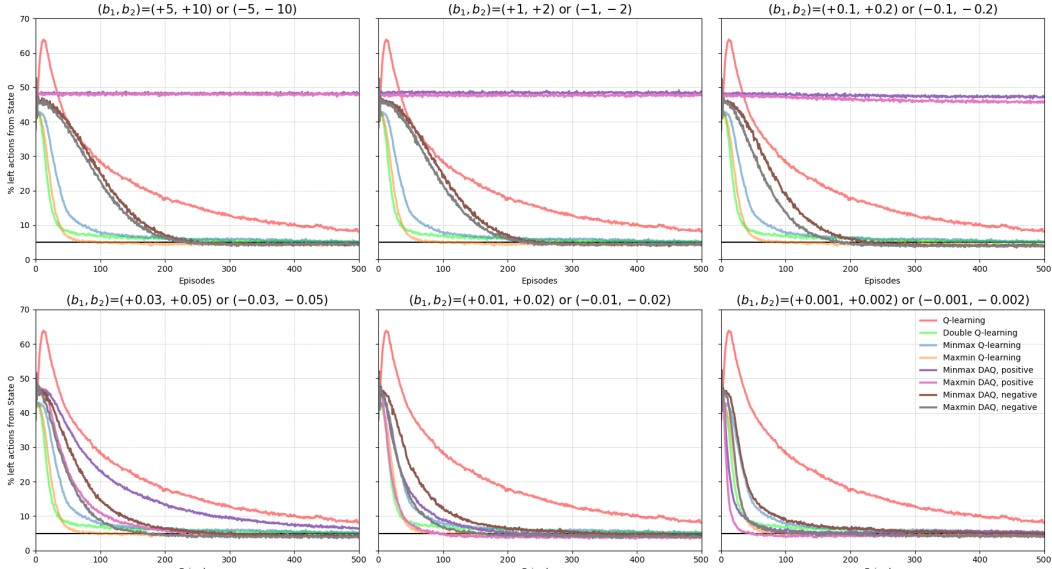

(a) Results with various shift values, which are presented in titles of each subfigures. $\alpha_n = \frac{10}{n+100}$ and $M = 8$ is used. The graphs of Q-learning, Double Q-learning, minmax Q-learning and maxmin Q-learning are the same for all six subfigures for better comparison. From upper left to bottom right, the amount of shifts decreases and the performance of DAQs gets better.

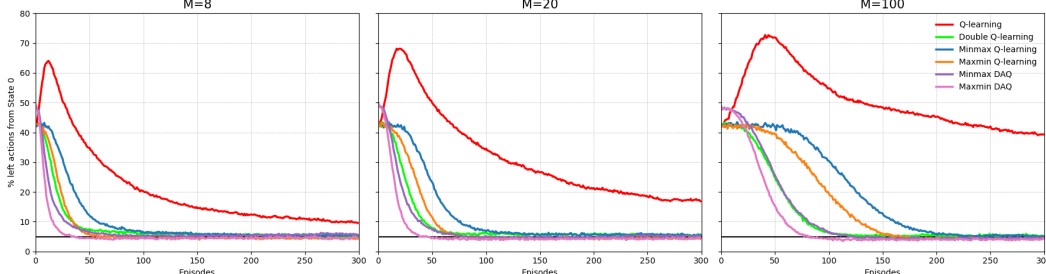

(b) Result using constant step-size $\alpha = 0.1$. From the first subfigure to the last, $M = 8, 20$ and $100$, respectively. $b_1 = +0.001$ and $b_2 = +0.002$ are used for both DAQs.

Figure 10: Result using Weng's Example. A 5% line is optimal. $\epsilon = 0.1$ and $\gamma = 1$ are used. Data is averaged over 10,000 experiments.

