# OpenReview forum: "Suppressing Overestimation in Q-Learning through Adversarial Behaviors"
_ICLR.cc/2024/Conference — ICLR 2024 Conference Withdrawn Submission_

### Official Review · Reviewer_KWyD · 2023-10-31

**Soundness:** 3 good
**Presentation:** 3 good
**Contribution:** 3 good
**Rating:** 6
**Confidence:** 4

**Summary:**

This paper proposes Dummy Adversarial Q-learning, a method to mitigate overestimation bias in Q-learning algorithms without altering the Q-learning objective. DAQ additionally aims to unify other overestimation methods, such as maxmin Q-learning and minmax Q-learning, the latter of which is also proposed in this paper. Experiments across standard RL domains show that both the maxmin and minmax DAQ variants achieve strong performance compared to standard Q-learning and double Q-learning.

**Strengths:**

The paper is generally very easy to follow, and generally has strong performance in the domains tested over. In particular, DAQ does very well compared to other baselines on sparse or negative reward tasks such as Sutton's example and Weng's example.

The theoretical claims the paper brings up are solid and seem correct.

**Weaknesses:**

I think the paper is missing some experiments that were done in the original minmax Q-learning paper, specifically those on the MinAtar and OpenAI Gym benchmarks. I think more experiments to show how well DAQ works in slightly more scaled-up environments would be a very strong addition to this paper. This is my main concern.

**Questions:**

I don't have any additional questions on this paper -- generally very solid.

**Details Of Ethics Concerns:**

None.

---

### Official Review · Reviewer_H3gW · 2023-10-31

**Soundness:** 2 fair
**Presentation:** 3 good
**Contribution:** 1 poor
**Rating:** 3
**Confidence:** 2

**Summary:**

The paper proposed an approach called dummy adversarial Q-learning to mitigate overestimation issue in bootstrap estimate. The basic idea is building upon the previous maxmin Q learning or minimax Q learning algorithms, with the addition of a constant, or random variable added to the reward for each Q function used. The authors also studied synchronous and asynchronous update of those Q functions. Empirical results are provided to show effectiveness.

**Strengths:**

1. The overestimation issue is important and is being studied by many researchers;

2. The paper presents its idea clearly.

**Weaknesses:**

The primary concern is in the significance and soundness.

Significance. The contribution is incremental. The proposed algorithm is a small addition to two existing methods. Although a small addition does not directly indicate a rejection, I do not see the significance of such addition.

Soundness. Note that the primary claim of “designing DAQ to mitigate overestimation” does not hold: the addition itself does not reduce overestimation, at least, the maxmin operator can already have the effect of reducing overestimation as shown by previous work. Simply adding a constant or r.v. (e.g. a zero mean gaussian) does not avoid overestimation. The benefits of such addition is also unclear: isn’t it possible adding a random noise could make the variance larger and hence slower down learning? The paper does not explain where the benefits of the proposed algorithm come from.

The synchronous and asynchronous updates do not align well with the goal of reducing overestimation.

Furthermore, the theoretical contribution about convergence is very intuitive.

**Questions:**

See above.

---

### Official Review · Reviewer_qny6 · 2023-11-01

**Soundness:** 2 fair
**Presentation:** 3 good
**Contribution:** 2 fair
**Rating:** 5
**Confidence:** 4

**Summary:**

This paper proposes Dummy Adversarial Q-Learning, a method for suppressing overestimation bias in Q-learning. The authors aim to address the commonly-known problem of overestimation of the Q-value in Q-learning due to the inner maximization in the update rule. Two algorithms are introduced: minmax Q-learning, which switches the order of max-min Q-learning from previous work. The second algorithm is Dummy Adversarial Q-Learning (DAQ). This approach adds a constant to the Q-value for each Q-value in the ensemble of Q-values.

**Strengths:**

+ The paper is written fairly clearly and the method is clearly described.
+ The experiments are quite comprehensive, at least in the toy domains that they are used in.
+ The finite-time convergence analysis is nice to see, and the proof seems correct.

**Weaknesses:**

+ I am not entirely convinced by the theoretical basis for the proposed method. In particular, in section 4.2 it's stated that the addition of $b_i$ doesn't change the optimal policy as it simply adds a constant bias to the Q-value. This is certainly true in the infinite-horizon case, but I don't think it's true in the finite-horizon case, which several (most?) of the experiments are set in. For concreteness, consider a very simple chain MDP like a <- b <-  c -> d where a and d are terminal states, the initial state is c, and the reward for each transition is -1. The optimal policy is obviously to go to d. But adding a reward of 2 to every transition means the optimal policy is to go to b then a. It seems that for the claim to work in the finite-horizon case, some special treatment of terminal states is necessary. However I can't see any mention of this in the paper. Perhaps the correctness of the approach still goes through, but it's not obvious to me.
+ The proposed change is relatively minor: a swapping of max and min and addition of a constant to the Q-value.
+ The experiments are relatively small-scale, mainly on toy examples. Examining previous work such as [Lan 2020], the proposed methods are usually evaluated on larger-scale problems where deep Q-learning is applied, such as the arcade learning environment. In addition, the improvements of this method over maxmin Q-learning seem relatively limited in the toy experiments that are examined.
+ The choice of the values for the bias terms $b_i$ seems somewhat arbitrary, and lack a principled foundation. As such, it's an additional hyperparameter that can be tuned to improve performance over the baselines.

**Questions:**

+ Does the correctness of DAQ still hold in the finite-horizon case? If not, is DAQ optimizing a different objective on the finite-horizon MDPs?
+ Do you have any experiments showing that DAQ maintains its good performance on harder RL problems such as atari, etc?
+ Is there a more principled method to choose the bias terms?

---

### Official Review · Reviewer_mxLo · 2023-11-09

**Soundness:** 1 poor
**Presentation:** 3 good
**Contribution:** 2 fair
**Rating:** 3
**Confidence:** 3

**Summary:**

The paper proposes the algorithm dummy adversarial Q-learning (DAQ), as a unification of minmax and maxmin optimization of Q value in reinforcement learning (RL). The major difference is to introduce a constant shift of reward for different Q estimators. This trick is quite straightforward, and shown to be effective in three small-scale RL tasks.

**Strengths:**

The paper is well written and easy to follow. The method is described clearly.

The discussion of the background literature provides a good motivation for the proposed method, as well as illustrating the connections.

The unification viewpoint of minmax and maxmin optimization for Q value is good, connecting them with two-player Markov games is reasonable.

**Weaknesses:**

The switching of maxmin operator to minmax Q-learning and adding reward shifts is quite straightforward.

The only difference of the minmax/maxmin DAQ with minmax/maxmin Q-learning seems to be the constant reward shift, which is a hyperparameter for performance tuning. The paper does not explain in detail about the choice of shift value, or any method for determining the shift value. What will the performances be affected if the shift values are different?

The two DAQ algorithms are proposed with $N$ Q estimators, but in experiments only $N=2$ are tested. This does not justify the claim that “A larger N might lead to slower convergence but has the potential to significantly mitigate over- estimation biases.” More experiments about the choice of $N$ should be added.

The theoretical analysis can be improved. First, the assumption of stationary state distribution at each time is quite strong, which is not usually satisfied in MDP. Second,
for Thm. 1, does the bound hold for both minmax and maxmin DAQ in exactly the same way? Please explain in the paragraph. Third, the disappearance of the first term in the bound with decreasing $\alpha$ is not theoretically justified. A decaying $\alpha$ requires further analysis.

For the experiments, please justify the choice of epsilon-greedy schedule and step size for each case. Especially for the Grid Wrold, please provide references or justification for the inverse square root epsilon and the choice of value 0.8 in step size.

The paper tries to address the overestimation of Q values, but no experiment directly shows the estimated  Q values compared with the ground truth. This comparison should be added to demonstrate the overestimation or underestimation and effects of the proposed methods.

Experiments on larger scale environments with longer horizons are expected to be evaluated. How does the usage of neural network approximation (e.g. DQN) affect the current methods?

**Questions:**

In Sec. 4.1 minmax Q-learning, does it require $i\neq j$?

Theoretically speaking, why is a constant reward shift sufficient for alleviating the overestimation bias? Is there a case requiring a state-wise reward shift?

Does switching values of $b_1, b_2$ affect the results? The order should not matter from Eq. (2-3), but do the optimal $b$ values depend on the initialization of Q values?

From the experiments on three tasks, maxmin DAQ seems to consistently outperform minmax DAQ, any explanation on this?